# Surgeon Dominated Design Can Improve the Accuracy of Patient-Specific Instruments in Kinematically Aligned TKA

**DOI:** 10.3390/jpm12081192

**Published:** 2022-07-22

**Authors:** Liang Wen, Zhiwei Wang, Desi Ma, Tiebing Qu

**Affiliations:** 1Department of Orthopeadics, Beijing Chaoyang Hospital, Capital Medical University, Beijing 100020, China; wenliang@ccmu.edu.cn (L.W.); madesi@126.com (D.M.); 2The Center of Diagnosis and Treatment for Joint Disease, China Rehabilitation Research Center, Beijing 100068, China; qtb@medmail.com.cn

**Keywords:** total knee arthroplasty, patient specific instrumentation, design, kinematic alignment, bone resection, accuracy

## Abstract

Precise bone resection is mandatory for kinematically aligned total knee arthroplasty (KA-TKA). This study is to investigate whether surgeon-dominated design can alter the accuracy of patient-specific instrumentation (PSI) in KA-TKA compared with the engineer design. A total of 24 patients (24 knees) who underwent KA-TKA in our institution were assigned to an engineer design group (10 knees) and surgeon design group (14 knees) chronologically. A novel portable medical-engineer interactive application can greatly enhance the surgeon’s participation in PSI design. The bone resection discrepancies were used to evaluate the accuracy of PSI in bone resection. The overall discrepancy of bone resection was reduced by surgeon-designed PSI compared to engineer-designed PSI by 0.33 mm. Surgeon-designed PSI seems to reduce the outliers in terms of relative discrepancies in bone resection as well, but it does not reach statistical significance. Moreover, surgeon-designed PSI could significantly improve the accuracy of PSI in the restoration of the joint line in terms of medial proximal tibial angle and mechanical lateral distal femoral angle. This study indicates that the dominance of surgeons in both PSI design and subsequent surgical operation should be emphasized in efforts to improve the accuracy of PSI.

## 1. Introduction

The alignment target of kinematically aligned total knee arthroplasty (KA-TKA) is to restore the articular geometry of tibia-femoral joint and the pre-arthritic joint laxity, while abandoning the neutral coronal alignment of mechanical alignment (MA) [1]. Many studies have shown that KA has advantages in restoring knee kinematics [2,3], improving patients’ satisfaction [4], reducing soft tissue disturbance [5], and a promising mid-term survivorship of implants [6].

As a three-dimensional alignment method, KA has strict requirements for bone cutting accuracy [4,7], a variety of computer-aided alignment tools, such as patient-specific instrumentation (PSI), navigation and even robotic assistant surgery, were used to improve the accuracy of bone resection and components implantation [8,9,10,11,12]. PSI plays the role of positioning tools or jigs, which help in executing surgeries with more accuracy, less operative time and minimal invasive purposes. Although PSI has been popularized in TKA for years [13], there is no consensus on its actual values in terms of accuracy, reliability or feasibility [14,15,16,17,18,19,20,21].

Many factors may impact the results of accuracy, including the error of the preoperative landmarking on the 3D model during the design stage, medial parapatellar approach, ambiguous positioning of the PSI on the cortex, sawing error, final impaction of components, the experience of the surgeons, the quality of medical images for postoperative assessment, and so on [22]. Only a few studies mentioned the role of surgeons in the PSI design process [23,24]. Since PSI was initially used for MA-TKA, an engineer involved in the design of KA PSI may have been more or less influenced by MA philosophy. The alignment target of KA-TKA is not in accordance with neutral coronal alignment, therefore in the early stages of KA PSI development, comprehensive participation of a surgeon in the design process of each PSI maybe more conducive to ensuring the accuracy. Therefore, the purpose of this study is to investigate whether surgeons’ participation in PSI design can improve its accuracy in KA-TKA.

## 2. Materials and Methods

### 2.1. Patients

In this retrospective study, a prospectively managed medical record database and corresponding medical imaging database were queried. A total of 24 consecutive patients (24 knees) undergoing primary KA-TKA assisted with PSI between November 2018 and December 2019 in a single institute were investigated. All patients were meticulously selected according to the following inclusion criteria: knee osteoarthritis (OA) with grade III or grade IV of Kellgren-Lawrence classification, medial proximal tibial angle (MPTA) ≥ 85° in varus knees, ≤5° of valgus knee with no medial collateral ligament (MCL) dysfunction or posterior condyles dysplasia. Exclusion criteria: inflammatory arthritis, previous collateral ligament or cruciate ligament rupture, previous intraarticular fracture, >20° of flexion contracture and genu recurvatum. Patients were divided into engineer group and surgeon group based on whether surgeons were involved in the PSI design. Patient demographics were summarized in Table 1.

### 2.2. Design and Manufacturing of PSI

In this study, the design and manufacturing of all PSIs were based on full-length computed tomography (CT, Slice thickness, 0.625 mm) of lower extremities. It was mandatory to compensate for the thickness of the articular cartilage by 2 mm in the CT-based PSI design. Articular surface-based PSI design philosophy was employed in this study [10].

On the femoral side, the distal and posterior facets of medial and lateral femoral condyles must be resurfaced by femoral components. The sizes of femoral components were determined according to the anterio-posterior dimension of femoral condyles. On the tibial side, the resected medial and lateral tibial plateau should be replaced by tibial component and polyethylene insert, rotational alignment of tibial component was determined by the connection line between the insertion of PCL and medial border of tibial tubercle, and posterior slope of tibial resection was in accordance with the natural posterior slope of medial plateau, the size of tibial component was determined by the best fit of virtual resection surface.

NX 9.0 (Siemens PLM Software, Plano, TX, USA) was used for the design of PSI (Figure 1). Rapid prototyping technology (Formiga P 110, EOS, Krailling, Germany) was used for 3D printing of the PSI. The printing material is medical nylon (PA2200 Polymer powder, EOS, Krailling, Germany).

### 2.3. Grouping of Patients

The design of the first 10 PSIs (engineer group) were accomplished by the engineers according to the above design philosophy. In the subsequent 14 knees (surgeon group), surgical operators comprehensively participated in the design of PSI through an application (EZguideTM, Naton Medical Technology Innovation Center, Beijing, China).

The advantage of this App is that the surgeon can adjust the prosthetic components in 6 degrees of freedom according to the technical requirements of different alignment (Figure 2). Moreover, the confirmed design protocol can be brought into the operating room via a portable device, making the intraoperative verification of the accuracy of PSI as easy as querying the patient’s radiograph through PACS integrated in operating room. Intraoperative verification is to confirm the consistency between the preoperative design and the intraoperative findings in terms of the geometry and thickness of the excised bone fragments and the shape of the bone resection surface (Figure 3).

### 2.4. Intraoperative Positioning of PSI

The CT-based PSI does not take account of the thickness of residual articular cartilage which should be removed by using a special curette before the PSI is secured to its unique position. In addition, the periosteum on the anterior aspect of the distal femur and on the medial-proximal aspect of the tibial tubercle should also be completely removed and all osteophytes should be kept in place so that the PSI can sit on the bony surface firmly and accurately. In this study, PSI was used for all cuts by positioning of the key locating pins instead of providing the slots for saw blade. After the 4 locating pins were in place (Figure 4a), the distal 2 pins and PSI were removed, and a specific cutting block was seated in line with the remaining locating pins for distal femoral resection (Figure 4b). Then a 4-in-1 cutting block was inserted into the distal pinholes for anterior and posterior condyle and chamfer resections. A similar procedure applied to the tibial side (Figure 4c), then a specific tibial cutting block was replaced for tibial resection (Figure 4d).

On the femoral side, before the securing of femoral PSI, compare the gap between PSI and print bone with that between PSI and the native distal femur (Figure 5). The geometry of this gap is very sensitive to the rotation of the PSI on the sagittal plane, and it is critical to reduce the incidence of less accuracy in posterior femoral condyle cuts, and such inaccuracy had been reported in a published literature [25]. On the tibial side, draw a line through tibial spine perpendicular to the articular surface of the tibial plateau on the full-length radiograph, then the radiograph with reference lines was brought into the operating room. Before the securing of tibial PSI, use the extra-medullary alignment rod affiliated with PSI to verify whether its orientation is consistent to that of the radiograph (Figure 6). Posterior cruciate-retained (CR) prostheses (Gemini MK II, Link, Hamburger, Germany) were used in this study.

### 2.5. Assessment of the Accuracy of PSI

The assessment of the accuracy of PSI mainly included two sets of parameters: the discrepancies of bone resection and the changes of joint line orientation. As far as the discrepancy of bone resection is concerned, the bone resection of each key facet was measured by caliper and recorded, and was compared with the thickness of components after compensation of cartilage (2 mm) and kerf of the saw blade (1.5 mm). This discrepancy is defined as positive when the actual resection thickness is larger than the component’s one, otherwise it is defined as negative. As far as the changes of joint line orientation are concerned, mechanical lateral distal femoral angle (mLDFA) and medial proximal tibial angle (MPTA) were used as assessment parameters. Compare the joint line parameters measured before and 3 months after the operation (Figure 7). Any intraoperative additional bone cut or ligament release, defined as out of plan manipulation, were recorded as well (Table 1).

### 2.6. Statistical Analysis

Data distribution of all measurement parameters was assessed by the Kolmogorov-Smirnov test. Normal distributed variables were presented as mean ± standard deviation, while attribute data were presented with absolute number. The independent samples *t* test was used to compare the variable data of patient demography between engineer group and surgeon group, General linear model and multivariate analysis of variance (ANOVA) were used to explore the effects of PSI types (engineer group, surgeon group), side (medial, lateral), and resection facets (distal femur, posterior condyle, tibial plateau) on the accuracy of bone cut. Similarly, the effects of PSI types and side (femoral side: mLDFA, tibial side: MPTA) on the restoration of joint line were investigated using multivariate analysis of variance. Fisher’s exact test was used to compare the outliers of bone resections between groups. Neither sample size calculation nor test power estimation was performed in the current study. All data analyses were conducted using SPSS (version 22.0, Chicago, IL, USA). The level of significance (*p* value) was set at 0.05.

## 3. Results

### 3.1. Descriptive Statistics of Patient Demographics

No significant difference in terms of age, BMI or operative time was shown between engineer group and surgeon group. Due to the small sample size of this case series, attribute data were only presented in absolute numbers (Table 1). All patients did not receive medial collateral ligament or lateral ligament structure releasing in our study. Four patients received lateral patellar retinaculum releasing, three patients received 2 mm of manual extra-cut on the medial tibia plateau, and +2 mm thicker liner was used in two patients.

### 3.2. Accuracy of Bone Resection

The effects of three independent variables, including PSI type, facets and side, on the bone resection difference were shown in Table 2. The results of multivariate ANOVA showed that the main effects of PSI type and facets on the difference of resection were significantly different (*p* < 0.001), while side had no significant effect on the difference of resection (*p* = 0.608). The gross difference of bone resection was reduced by surgeon-designed PSI compared to engineer-designed PSI by 0.33 mm (*p* < 0.001). The bone resection differences of two PSI types in different facets were shown in Figure 8a–c. In the subsequent pairwise comparisons of different facets, the resection of tibial plateau was found 0.43 mm thinner than that of the distal femur (*p* < 0.001), and 0.68 mm thinner than that of the posterior condyle on average (*p* < 0.001). The resection of posterior condyle was 0.25 mm thicker than that of the distal femur on average (*p* = 0.029). Moreover, surgeon-designed PSI seems to reduce the outliers in terms of relative discrepancies in bone resection (Figure 8a–c), but did not reach the statistically significant level (5/55 vs. 2/82 facets, *p* = 0.128).

### 3.3. Accuracy of Joint Line Restoration

When the effects of different PSI types on the restoration of joint line were investigated, absolute deviations were used as the dependent variables. The results of multivariate ANOVA demonstrate that surgeon-designed PSI significantly improves the gross accuracy in the restoration of the joint line (*p* = 0.01), but no significant difference was found between the femoral side and the tibial side (*p* = 0.466) (Table 3, Figure 7).

## 4. Discussion

The results of this study demonstrate that surgeon’s participation in preoperative design can significantly improve the accuracy of PSI in terms of bone cutting accuracy and joint line reconstruction in the early stages of developing PSI-KA. In this study, a self-developed surgeon-engineer interactive application allows the surgeon to fully participate in the design of PSI. The surgeons have the authority to adjust all design parameters, including adjusting the position of the femoral component and tibial component in six degrees of freedom, and change the size of the components. Thanks to the easy-to-use interface of this application and efficient cloud-based data exchange, it usually only takes 10 to 15 min for the surgeon to complete the adjustment of the PSI design parameters. This application combined with intraoperative verification has significantly enhanced the accuracy of PSI in KA-TKA. Moreover, the sizes of implanted components matched up with the preoperative designed ones in all knees.

In the design of PSI, what role the doctor should play is an interesting topic. A survey demonstrated that although most surgeons have shown great interest in the application of PSI in TKA, 47% of surgeons believed that it was the manufacturer’s fault if PSI is not accurately aligned in the operation [26]. However, the absence of the participation of the surgeon may be one of the important reasons for the inconsistent results of PSI accuracy. Pietsch et al. found that surgeon’s participation in preoperative PSI design could make preoperative planning more accurate and significantly reduce the intraoperative adjustments [24], but only a few design parameters, including femoral flexion, component sizes and tibial rotation, were allowed to be adjusted by the surgeons in their study.

The accuracy of PSI is also affected by other factors. For example, PSI designs are based on a three-dimensional graph model, which are obtained by computed tomography (CT) or magnetic resonance imaging (MR) reconstruction. Although many studies have shown that MR-based PSI is more accurate than a CT-based one [27,28], the latter has many advantages, such as quick access, cheap, automatic software-based reconstruction, etc., and more importantly, ease to obtain thinner slices (0.625 mm in this study), which is critical for reproducing the anatomical articular details. After weighing the pros and cons, CT-based PSI with better cost-effectiveness was applied in this study.

Other studies dedicated to increasing the accuracy of PSI are mainly to improve the design, especially the design of the tibial PSI [29,30]. A large number of studies have confirmed that PSI has advantages in improving the alignment of the femoral component and global mechanical alignment, but with an increased risk of outlier for the tibial component alignment [15,31,32]. In the current study, in order to improve the accuracy of the tibial PSI, an extended cannula to accommodate alignment rod was employed to verify the accuracy of the PSI installation as per Yamamura’s method [30]. Even so, the results of this study found that the accuracy of the tibial cut is slightly inferior to that of the femoral cut, which is mainly manifested in the greater dispersion and more outliers of the tibial cut data. Surprisingly, surgeon’s participation in the preoperative PSI design can further reduce such tibial cut errors, and it has certain advantages in reducing outliers. This may be attributed to the visual surgeon-engineer interactive application, which can assist surgeons in executing intraoperative bone cut verification. The highlighted bone cut fragments on the application interface makes it easy to identify outliers when using angel wing to double check.

Some results of this study need additional explanation. The PSI used in this study showed the tendency of a slightly thicker posterior condylar resection and distal femoral resection, while showing a tendency of a slightly thinner proximal tibial resection. This tendency is not from systematic errors, but we deliberately performed it in the PSI design process. Because the polyethylene insert of the Gemini CR prosthesis is a deep dish-like design with an elevated posterior edge, if the posterior femoral condyles were equally replaced, the flexion gap would be too tight.

There were several limitations in this study. First, the current study is not a randomized controlled design. The objects are enrolled into engineer group and surgeon group according to the chronological sequence of receiving TKA. Therefore, it is reasonable that the learning curve of PSI would affect the results of this study. Because our research group had used the exact same commercial PSI for many years, the adverse effects of the learning curve could be significantly diminished. Secondly, the sample size of this study is small, but this study can still draw the conclusion that surgeon’s participation in the design can significantly improve the accuracy of PSI.

## 5. Conclusions

The contribution of the surgeons in the design process of PSI was strengthened further in this study. If a surgeon dominated the design process of PSI and surgical operation one after another, the intraoperative positioning and fixation of PSI would be more in line with the preoperative planning, so the accuracy of PSI could be significantly improved as well. We hope such a result will serve as a reference for those about to perform PSI KA, especially at their learning curve stage. A portable surgeon-engineer interactive application makes it more feasible for surgeons to participate in PSI design and intraoperative bone cut verification.

## Figures and Tables

**Figure 1 jpm-12-01192-f001:**
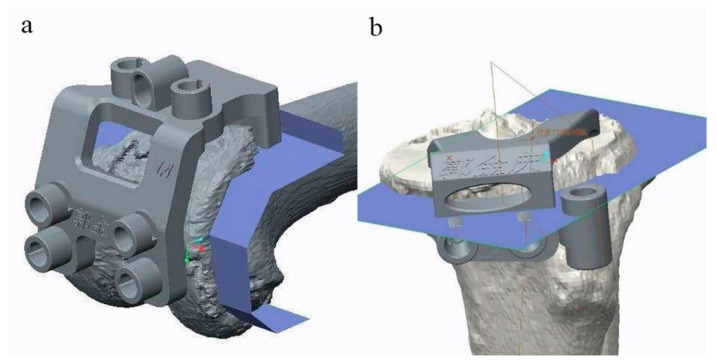
Screenshots of the femoral (**a**) and tibial (**b**) patient-specific instrumentation (PSI) design. Note: The embossed Chinese on the PSI design prototype is the name of the corresponding patient.

**Figure 2 jpm-12-01192-f002:**
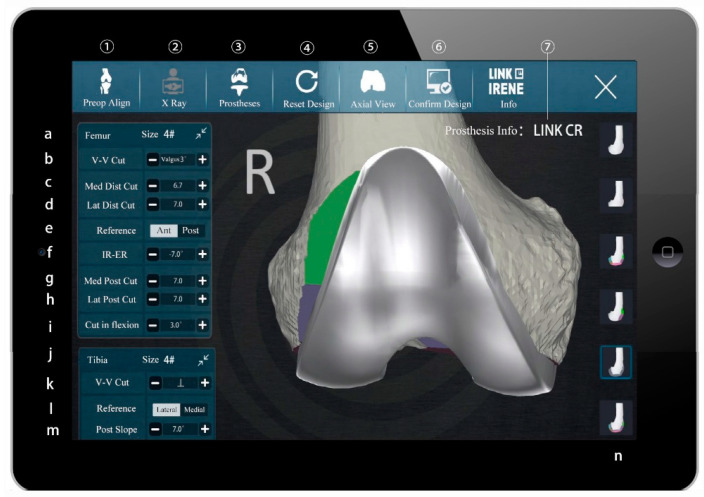
The main working page of EZguide^TM^ application. The interactive buttons in the left sidebar: a femoral component sizing, b Varus-Valgus resection angle of the distal femur with reference to the mechanical axis of the femur (deg.), c resection of medial distal femoral condyle (mm), d resection of lateral distal femoral condyle (mm), e anterior or posterior referencing, f femoral Internal-External rotation angle in axial plane with reference to trans-epicondylar axis (deg.), g resection of medial posterior femoral condyle (mm), h resection of lateral posterior femoral condyle (mm), i flexion angle of femoral component in sagittal plane with reference to the mechanical axis of the femur (deg.), j tibial component sizing, k Varus-Valgus resection of proximal tibia on the coronal plane with reference with the mechanical axis of tibia (deg.), l medial or lateral tibial resection reference, m tibial posterior slope (deg.) and the resection depth of tibial resection (at the bottom of main page, not displayed on the screen, mm). The interactive buttons on the top of screen: ① Preoperative alignment of lower extremities, ② Preoperative radiographs, ③ Available commercial prostheses, ④ Reset the design, ⑤ Axial view, ⑥ Confirm the design, ⑦ Selected prosthesis information. The buttons in the right sidebar provides different visualization options, including intact bone anatomy with key landmarks, the geometry of resection surface, the resected bones in different colors, the coverage of resection surface by prosthesis, and the superimposed bone and prosthesis.

**Figure 3 jpm-12-01192-f003:**
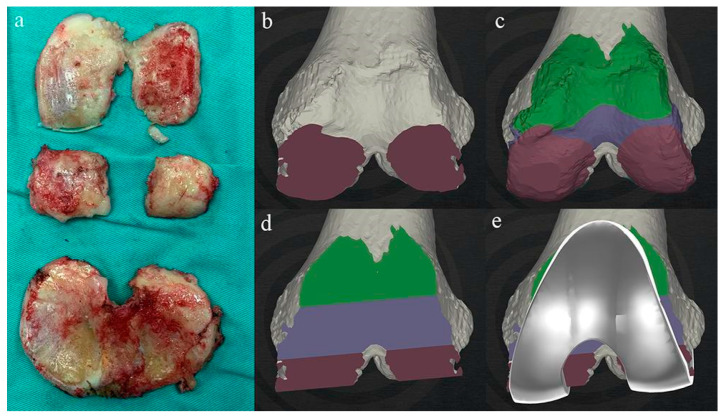
Intraoperative verification of bone resection. The geometry and thickness of resected bone fragments (**a**) were used to compare the shape of bone cut surface (**b**,**d**), the geometry of virtual bone cut fragments (**c**), the thickness of component trials, and the coverage of the bone cut surface (**e**).

**Figure 4 jpm-12-01192-f004:**
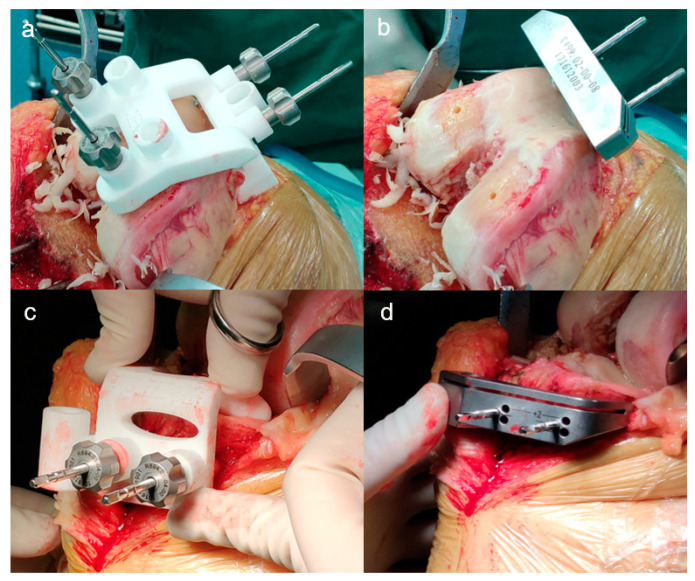
PSI was used for positioning of the key locating pins intraoperatively. (**a**) Two pairs of locating pins were drilled through the cannulas integrated into the femoral PSI. (**b**) The distal pins and PSI were removed, and a distal femur cutting block was seated in line with the remaining pins for distal bone resection. The two pin holes at the distal condyle were used for positioning of the 4-in-1 cutting block. (**c**) Locating pins were drilled through the cannulas integrated into the tibial PSI. (**d**) Switching tibial cutting block for tibial bone resection.

**Figure 5 jpm-12-01192-f005:**
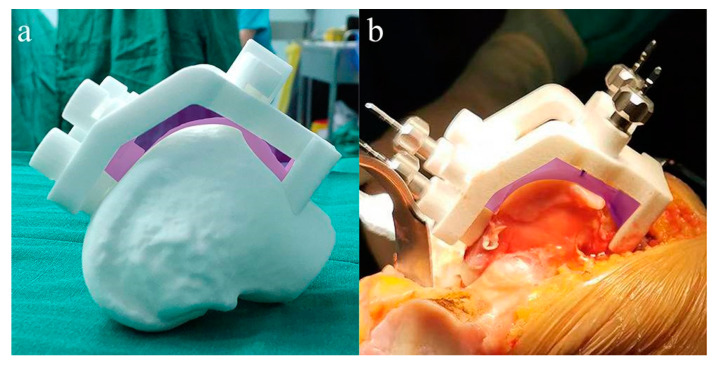
Comparison the gap between PSI and the printed bone (**a**) with that between PSI and native femur (**b**) is critical for the accuracy on the sagittal plane. The gaps were highlighted with translucent pink shadows.

**Figure 6 jpm-12-01192-f006:**
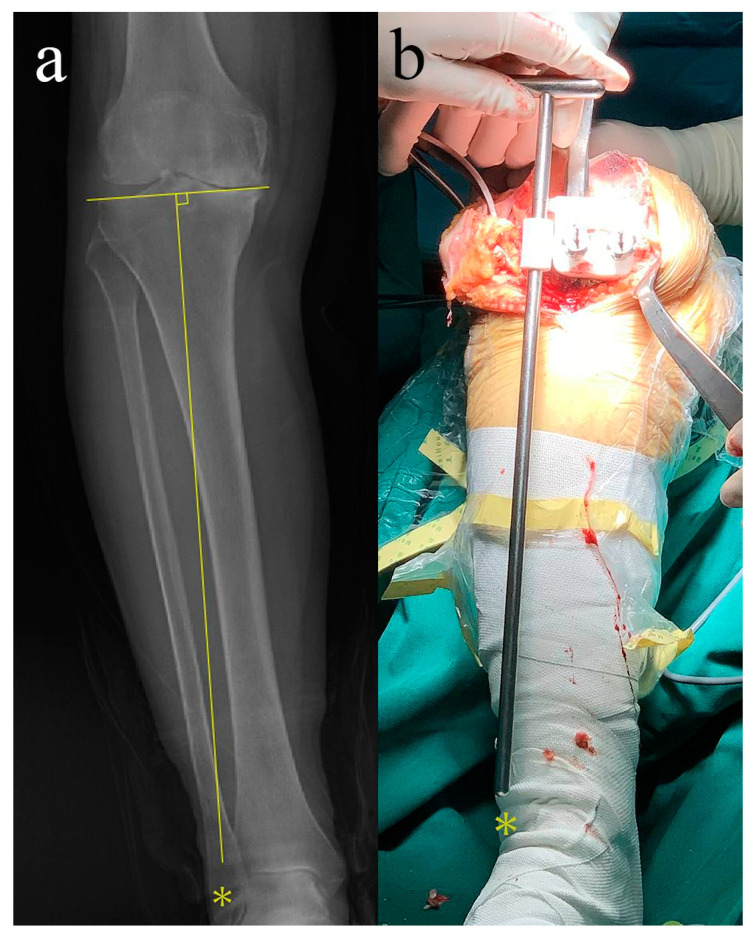
Intraoperative verification of the accuracy of the tibial PSI on the coronal plane. The yellow stars represent the orientation of the vertical line of the tibial articular surface. (**a**) Determining the orientation of the vertical line of the tibial articular surface in preoperative X-ray. (**b**) Conforming the orientation of the vertical line of the tibial cutting surface during the surgery.

**Figure 7 jpm-12-01192-f007:**
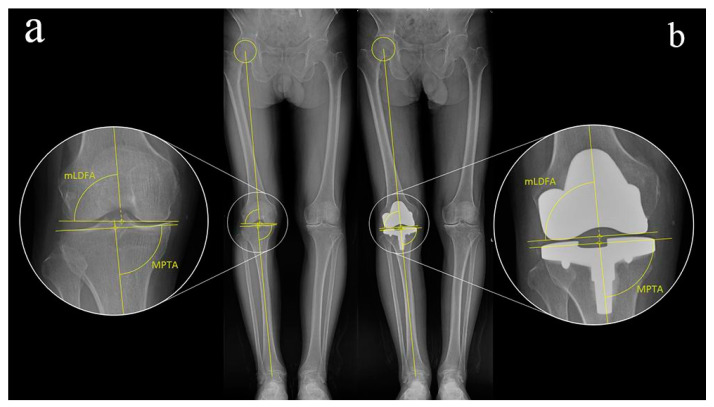
Measurement of the preoperative (**a**) and postoperative (**b**) joint line orientation parameters. mLDFA mechanical lateral distal femoral angle, MPTA medial proximal tibial angle.

**Figure 8 jpm-12-01192-f008:**
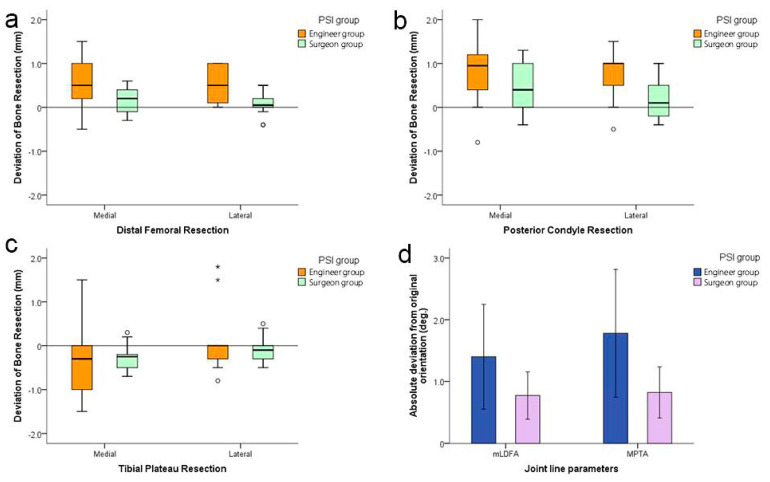
The effects of doctor-designed PSI on bone resection accuracy (**a**–**c**) and joint line restoration (**d**). Note: Asterisks and circles in plots represent outliers.

**Table 1 jpm-12-01192-t001:** Preoperative patient demographics and other parameters of engineer-designed PSI group and surgeon-designed PSI group.

	Engineer Group (*n* = 10)	Surgeon Group (*n* = 14)	t	Sig.
Age (years)	66.8 ± 6.5	67.1 ± 4.1	−0.126	0.901
Side (Left/Right)	6/4	8/6	NA	NA
Gender (Male/Female)	5/5	4/10	NA	NA
Height (cm)	165.5 ± 8.1	163.7 ± 5.8	0.631	0.534
Weight (kg)	72.1 ± 13.9	67.6 ± 11.1	0.886	0.385
BMI (kg/m^2^)	25.1 ± 3.2	25.1 ± 2.6	0.892	0.382
Preop. Alignment (Varus/Valgus)	9/1	14/0	NA	NA
OP time (min)	78.8 ± 9.9	75.9 ± 10.9	0.677	0.505
Femoral component size (#2/#3/#4/#5)	5/2/2/1	8/4/2/0	NA	NA
Tibial component size (#2/#3/#4/#5)	5/2/2/1	8/4/2/0	NA	NA
Thicker (+2 mm) liner (Y/N)	1/9	1/13	NA	NA
Lateral patellar retinaculum release (Y/N)	2/8	2/12	NA	NA
Intraoperative tibial extracut (Y/N)	2 medial side/8	1 medial side/13	NA	NA

PSI patient-specific instrumentation, BMI body mass index, NA not applicable, OP time operative time, Y yes, N no.

**Table 2 jpm-12-01192-t002:** Bone resection discrepancies (mean ± SD) between the preoperative PSI design and intraoperative actual measurements.

PSI Group	Facet	Distal Femur (mm)		Post. Condyle (mm)		Tibial Plateau (mm)		AVONA			
	Side	Medial	Lateral	Medial	Lateral	Medial	Lateral	Main Effect	*p* Value	Interaction	*p* Value
Engineer group		0.5 ± 0.6	0.5 ± 0.4	0.8 ± 0.8	0.8 ± 0.6	−0.28 ± 0.9	0.2 ± 0.8	PSI Facets Side	<0.001 <0.001 0.608	PSI * Facet PSI * Side Facet * Side PSI * Facet * Side	0.325 0.326 0.101 0.925
Surgeon group		0.1 ± 0.3	0.1 ± 0.3	0.4 ± 0.5	0.2 ± 0.5	−0.3 ± 0.3	−0.1 ± 0.3				

PSI patient-specific instrumentation, Post. condyle, femoral posterior condyle.

**Table 3 jpm-12-01192-t003:** The effects of PSI on the absolute deviation of joint line orientation (mean±SD).

PSI Group	Side of Joint Line (deg.)		ANOVA (*p* Value)		
	mLDFA	MPTA	PSI	Side	PSI * Parameters
Engineer group	1.40 ± 1.18	1.78 ± 1.44	0.010	0.466	0.576
Surgeon group	0.77 ± 0.66	0.82 ± 0.71			

PSI patient-specific instrumentation, mLDFA mechanical lateral distal femoral angle, MPTA medial proximal tibial angle. Note: * represents interaction.

## Data Availability

The data presented in this study are available on request from the corresponding author.

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
