# Peer review of "Surgeon Dominated Design Can Improve the Accuracy of Patient-Specific Instruments in Kinematically Aligned TKA"

_jpm, 2022, doi:10.3390/jpm12081192_

Round 1

Reviewer 1 Report

Wen et al., have submitted a manuscript describing the design strategies involved in total knee arthroplasty. Overall, the manuscript is well written and covers an important area of research, concluding that surgeon-dominated designs are better in TKA. I have the following suggestions for authors to consider in their revised manuscript. 

-Can the authors revise the working guide in fig.2 in English, so it can be easily accessible by other researchers?

-Please define in the introduction what is meant by “engineer group” and “surgeon group”

Author Response

Response to Reviewer 1 Comments

Point 1: Can the authors revise the working guide in fig.2 in English, so it can be easily accessible by other researchers?

Response 1: Thank you for your suggestion. Figure 2 has been translated into English and the annotation for Figure 2 has also been modified accordingly in the manuscript (Line 119-137).

Point 2: Please define in the introduction what is meant by “engineer group” and “surgeon group”

Response 2: Thanks for the remind, it is very logical. Before "engineer group" and "surgeon group" appeared for the first time in Table 1 in the original manuscript, descriptions and definitions for "engineer group" and "surgeon group" were indeed missing. We have added the definitions "Patients were divided into engineer group and surgeon group based on whether surgeons were involved in the PSI design" in the manuscript (Line 64-65).

Thank you again for the above suggestions which highly improved the legibility and logic of the article.

Reviewer 2 Report

Dear authors

I thank you for submitting to JCM your work. 

Although your efforts I am concerned of the small to zero clinical relevance of your paper. The conclusion implied that surgeons must be in charge of another step (design PSI) which will increase the daily workload. The experience of the engineers is not taken into account. In my personal experience plus some colleagues experience who i specifically questioned about, the presence of an experienced engineer is of great help during RA-TKA or PSI design and PSI TKA execution. 

Author Response

Response to Reviewer 2 Comments

Point 1: The conclusion implied that surgeons must be in charge of another step (design PSI) which will increase the daily workload.

Response 1: Thanks for your comments and concern about the workload increase which is really a problem for us, but only for the very beginning. Our surgeons' workload in each PSI design was successfully controlled within 10 to 15 minutes in our practice after a couple of cases. In order to illustrate these situations, we have also added descriptive notes in the discussion section (Line 348-350).

Point 2: The experience of the engineers is not taken into account. In my personal experience plus some colleagues’ experience who I specifically questioned about, the presence of an experienced engineer is of great help during RA-TKA or PSI design and PSI TKA execution.

Response 2: Thank you for your comments about emphasizing the role of engineers which we totally agree with. There are also a lot of experienced engineers in our nation, they played an important role and gave us great help in the design of Robotic MA TKA and PSI MA TKA. KA, however, is a totally different story for them. In order to clarify the actual situation in our country, we have also added descriptive notes in the introduction section (Line 46-50).

Thank you very much indeed for such sincere and earnest comments which allow us to reconsider the clinical relevance of this study. KA, as a promising technology, is still in its infancy in our nation. In general, surgeons are often ahead of engineers in understanding KA, our research demonstrated that surgeon's participation in preoperative design can significantly improve the accuracy of PSI in terms of bone cutting accuracy and joint line reconstruction in the early stages of developing PSI-KA. We hope the result in this study will serve as a reference for those about to develop this technology, thus facilitating the collaboration between surgeons and engineers, which is necessary for further development and maturation of PSI-KA, although still a long way off. In order to illustrate these prospective, we have also added descriptive notes in the conclusion section (Line 416-417).

Reviewer 3 Report

Dear Authors:

I would like to congratulate with You for the paper. I would like You to better specify the surgical technique: were PSI-masks used just for initial cuts and later on guide-masks from the prosthesis manufacturer were used, or were PSI-masks used for all cuts? Also, I recommend minor English revision (line 123: does not better than doesn't, for example).

best regards,

Author Response

Response to Reviewer 3 Comments

Point 1: I would like You to better specify the surgical technique: were PSI-masks used just for initial cuts and later on guide-masks from the prosthesis manufacturer were used, or were PSI-masks used for all cuts?

Response 1: Thank you for such a constructive suggestion in terms of surgical technique which was not described clearly in original manuscript and should be specified in revision. A new Figure 4 was added (Line 157-163) and text description was amended (Line 177-244) to specify the surgical technique. Actually, in this study, PSI was used for all cuts by positioning of the key locating pins instead of providing the slots for saw blade. After 4 pins were fixed through the guide integrated in the femoral PSI (Figure 4a), 2 distal pins and PSI was removed, and a distal femur cutting block was seated in line with the remaining pins for distal bone resection (Figure 4b). Then, a 4-in-1 cutting block was inserted into the pinholes left on the distal cutting surface for anterior and posterior condylar and chamfer resections. The similar procedure was performed in the tibial side (Figure 4c), and switching tibial cutting for tibial resection (Figure 4d).

Point 2: I recommend minor English revision (line 123: does not better than doesn't, for example).

Response 2: Thanks for the tips and the amendments made in line 152.

Thanks a million for the above suggestions, which really made the details clearer and more standard.

Round 2

Reviewer 2 Report

dear authors

thanks for your clarifications